# Dendritic Cells and *Cryptosporidium*: From Recognition to Restriction

**DOI:** 10.3390/microorganisms11041056

**Published:** 2023-04-18

**Authors:** Iti Saraav, L. David Sibley

**Affiliations:** Department of Molecular Microbiology, Washington University School of Medicine, St. Louis, MO 63110, USA

**Keywords:** dendritic cells, *Cryptosporidium*, immune response, innate immunity, intestinal parasite

## Abstract

Host immune responses are required for the efficient control of cryptosporidiosis. Immunity against *Cryptosporidium* infection has been best studied in mice, where it is mediated by both innate and adaptive immune responses. Dendritic cells are the key link between innate and adaptive immunity and participate in the defense against *Cryptosporidium* infection. While the effector mechanism varies, both humans and mice rely on dendritic cells for sensing parasites and restricting infection. Recently, the use of mouse-adapted strains *C. parvum* and mouse-specific strain *C. tyzzeri* have provided tractable systems to study the role of dendritic cells in mice against this parasite. In this review, we provide an overview of recent advances in innate immunity acting during infection with *Cryptosporidium* with a major focus on the role of dendritic cells in the intestinal mucosa. Further work is required to understand the role of dendritic cells in the activation of T cells and to explore associated molecular mechanisms. The identification of *Cryptosporidium* antigen involved in the activation of Toll-like receptor signaling in dendritic cells during infection is also a matter of future study. The in-depth knowledge of immune responses in cryptosporidiosis will help develop targeted prophylactic and therapeutic interventions.

## 1. Introduction

*Cryptosporidium* is an intestinal protozoan parasite that causes severe diarrhea in human infants and young ruminants [1]. It is the leading cause of diarrheal-related death in children under 5 years of age worldwide [1,2]. According to the Global Enteric Multicenter Study, middle- and low-income countries carry the primary burden of cryptosporidiosis, which is the second leading cause of diarrheal episodes in children under 2 years of age [3,4]. Children presenting with the symptomatic disease are more likely to suffer from malnutrition and lower height and weight per age, and these deficiencies persist for years after the primary infection resolves. Surprisingly, even asymptomatic cases are associated with malnutrition and failure to thrive for several years beyond the initial infection [2]. Based on the different degree of infectivity for hosts, several species of *Cryptosporidium* exists, among which *Cryptosporidium parvum* (*C. parvum*) and *C. hominis* are known to be responsible for the most clinically significant infections in humans around the world [5]. *C. parvum* is zoonotic and infective to a broad host range, including animals and humans, while *C. hominis* is the primary cause of human-to-human infection [6,7]. Infection is typically acquired when a person consumes contaminated water or food, the spore-like oocysts travel through the gastrointestinal tract, excyst, and release motile stages as sporozoites. Then, sporozoites adhere to the surface of the host cell membrane, induce the formation of a parasitophorous vacuole, and invade the epithelium of the small intestine [1,8]. The exact mechanism through which sporozoites attach to the apical membrane of host epithelial cells (ECs) is still unknown but is thought to involve mucins that function as adhesins [9,10]. Within ECs, *Cryptosporidium* undergoes several rounds of asexual followed by sexual replication, which gives rise to new oocysts. Oocysts are shed in high numbers, and they are highly infectious and environmentally stable, leading to transmission [11]. The impervious oocyst wall protects the parasites from chemical disinfectants used in water treatment facilities, such as chlorine and which is why even in countries with advanced water treatment, the spread of infection is common [12].

*Cryptosporidium* is an intracellular parasite, but it remains extra-cytoplasmic without resulting in systemic infections [13,14]. However, the infection may extend to the biliary tract epithelium in patients with acquired immune deficiency syndrome-causing cholangiopathy, an important biliary disorder [1,15]. Cryptosporidiosis is associated with histological abnormalities in the crypt and villous structure, the destruction of brush border microvilli, and cellular infiltration, which results in serious nutritional deficiencies, dehydration, and diarrhea [16,17]. Immunity against *Cryptosporidium* infection in mice and humans is mediated by both innate and adaptive immune responses; however, immune responses often differ between mice and humans [18,19]. The innate immune response has an important role in controlling the infection by restricting the growth of the parasite during the early acute phase and initiating the adaptive immune response. The innate immune system in both mice and humans involves intestinal ECs, innate immune cells lying underneath them such as natural killer cells, dendritic cells (DCs), macrophages, chemokines, Toll-like receptors (TLRs), nucleotide-binding oligomerization domain-like receptors, and cytokines such as IL-18, TNF-α, and IFN-λ [18,19,20,21,22]. An initial response to *Cryptosporidium* infection primarily involves the communication between intestinal ECs and specialized innate immune cells. Besides being a physical barrier, ECs also act against *C. parvum* infection in mice by producing chemokines to attract immune cells at the site of infection [21,23]. Studies in human intestinal ECs show that chemokines such as CCL-5, CXCL-8, and CXCL-10 are upregulated in *Cryptosporidium* infection, and CXCL-8 is detectable in the stool samples of children infected with cryptosporidiosis [18,24]. Furthermore, ECs participate in both mouse and human defense by inducing apoptosis [25,26], increasing prostaglandin E2 production [27,28], and releasing antimicrobial peptides such as β-defensin-1, β-defensin-2, and LL-37 [29,30,31]. This parasite has developed several evasion strategies to avoid the host immune system’s killing effect, which is discussed in detail elsewhere [20,32].

The major mediator of resistance to *C. parvum* infection is IFN-γ, a key proinflammatory cytokine, and as a consequence, its neutralization or genetic depletion makes mice more susceptible to infection [33,34,35]. IFN-γ produced during *C. parvum* infection activates the STAT-1 pathway in human and mouse intestinal ECs to induce the expression of interferon-stimulated genes (ISGs) that exert several antiparasitic effector functions. One of the early studies using human enterocyte cell lines concluded that IFN-γ can inhibit *C. parvum* infection, but inhibitory activities in vitro are modest, and the mechanisms responsible for parasite restriction are not clear [36]. A recent study with *C. parvum* in mice revealed that intestinal type-I innate lymphoid cells are a major source of early IFN-γ, and IFN-γ-induced STAT1-mediated activity is uniquely required in enterocytes to limit parasite growth [37]. Furthermore, infection resulted in the upregulation of ISGs such as indolamine dioxygenase, guanylate binding proteins (GBPs), and immunity-related GTPases (IRGs) in enterocytes, but deletion experiments identified only the role of IRGs in the IFN-γ-mediated parasite control [37]. Studies of previously infected immunocompetent individuals showed that IFN-γ is produced from peripheral blood mononuclear cells upon ex vivo stimulation with a *C. parvum* crude extract in cryptosporidiosis patients [38]. The expression of IFN-γ is significantly correlated with the presence of *C. parvum* antibody and the absence of oocyst shedding [39]. Adaptive immunity depends on the function of T-cell-based cellular and B-cell-based humoral immune responses [40]. Although both of them are involved, T-cell immunity plays a major role in both mouse and human defense against *C. parvum* [18,41,42]. T-cell-deficient mouse models have established a strong protective role of adaptive immunity in the resolution of infection with *C. parvum*, specifically CD4+ T cells, whose role in the clearance of parasites has been recently confirmed in the natural mouse model *C. tyzzeri* [43]. Studies of infected human volunteers and chronically infected AIDS patients show that IFN-γ mRNA is detected in the jejunal biopsies of human volunteers post-infection but not in biopsies from patients with AIDS-associated cryptosporidiosis [39]. Human CD8+ T cells secrete IFN-γ early in infection and clear *C. parvum* from the infected intestine via cytotoxicity by lysing the infected intestinal ECs [44]. Although adaptive immunity is important for the complete clearance of the parasite, innate immune responses are critical in reducing the parasite burden during the initial stages. Additionally, despite evidence for the role of IFN-γ in response to infection, patients with inborn errors in the interferon pathway do not suffer more frequent or severe infection with Cryptosporidium spp. [45], suggesting there are alternative immune control mechanisms. With newly developed models and transgenic lines, our knowledge of early mucosal immune response to *Cryptosporidium* infection is constantly improving [43,46]. In this review, we provide an overview of recent advances in innate immunity acting during infection with *Cryptosporidium*, with a major focus on the role of DCs in the intestinal mucosa.

## 2. Dendritic Cells

Dendritic cells were first discovered by Paul Langerhans in 1868, who was a medical student when he described dendritically shaped non-pigmentary cells in the epidermis [47]. After almost 100 years, in 1973, Steinman and Cohn described a semi-adherent cell population in the spleen of mice as DCs with stellate shape and established that DCs are different from monocytes and macrophages in their appearance and behavior [48]. Currently, DCs are the most potent professional antigen-presenting cells that serve as a key link between innate and adaptive immune responses against pathogens [49]. DCs are a group of heterogeneous cell populations whose subsets differ in the expression of unique molecules, ontogeny, localization, and migration (Table 1). DCs perform different functions depending on the type of immune signals they receive from other cells while surveying the environment [50]. The development of DCs is a multistep differentiation process that begins in the bone marrow. The precursors of DCs leave the bone marrow and, through blood circulation, localize to the lymphoid and non-lymphoid tissues [51]. These subsets are identified as classical/conventional type-I DCs, conventional type-II DCs (cDC2), plasmacytoid DCs (pDCs), and monocyte-derived (Mo-DCs) [52,53]. Based on transcriptomic profiling, the most recent refined classification system confirmed these four subsets (cDC1, cDC2, pDCs, and Mo-DCs), with the introduction of a new layer of heterogeneity in cDC2 and pDCs subsets [54]. Most DCs are generated from a common precursor of cDCs and pDCs. The development of pDC continues in the bone marrow, while cDCs differentiate in the periphery [55]. During pathogen-induced inflammation, Mo-DCs are generated with DC-like properties via monocyte differentiation [53]. A number of transcription factors are known that control the specification and survival of DCs, and these factors change between subtypes. The differentiation of DCs from cDC precursors is based on key transcription factors, such as BATF3 (basic leucine zipper ATF-like transcription factor), interferon regulatory factor (IRF8), ID2, ZFBTB46 (zinc finger and BTB domain containing), the FLT3L growth factor, and granulocyte–macrophage colony-stimulating factor (GM-CSF). Specifically, the development of cDC1 is dependent on BATF3 and IRF8 [56]; cDC2 development is regulated by several transcription factors, including RelB, IRF2, and IRF-4 [52]; and the development of pDCs is dependent on E-protein transcription factor E2-2, IRF4, and IRF7 [57]. Immature DCs (iDCs) are less capable of migration and cytokine secretion, and they express lower levels of major histocompatibility complex (MHC) I and II and T-cell co-stimulatory ligands such as CD80 and CD86. However, they are equipped with fully functional endocytic machinery for antigen uptake. Once iDCs recognize a foreign antigen, the maturation process starts, resulting in enhanced migratory properties in these cells due to the increased expression of chemokine-homing receptor CCR7. Mature DCs reside mostly in the secondary lymphoid tissues, where they act as antigen-presenting cells (APCs) by extending their long dendrites [58]. DCs have the capacity to recognize different molecules on the surface of pathogens and phagocytose them. Once internalized, DCs process antigens for presentation to naïve T cells. DCs have the unique property to migrate to regional lymph nodes (LNs), where they activate naïve T cells, thereby initiating an adaptive immune response [59].

## 3. Interaction of Dendritic Cells with *Cryptosporidium*

*Cryptosporidium* replicates within a parasitophorous vacuole near the apical surface of the intestinal epithelium but does not invade deeper into the host cell cytoplasm [14]. *C. parvum* does not infect laboratory mice well, but immunocompromised mice are susceptible, and they have been used to glean insight into some of the features important in controlling infection. The recruitment of DCs at the site of infection by the chemokines released from infected intestinal ECs is an important early step in the recognition of this parasite (Figure 1). The importance of chemokine production in stimulating recruitment is evidenced by the increased susceptibility of chemokine-receptor-deficient mouse strains to infection [60]. In the neonatal mouse model, intestinal ECs produce a broad range of chemokines belonging to C, C–C, and C–X–C chemokine families in response to *C. parvum* infection [19,23,61]. Among these chemokines, the CXCL10 produced during infection by intestinal ECs is associated with the recruitment of DCs, resulting in a protective immune response in neonatal mice [61]. IFN-γ is important in inducing the proinflammatory chemokine CXCL10, as revealed by decreased DC recruitment in IFN-γ knockout (KO) neonatal mice infected with *C. parvum* [23]. Similarly, in humans, high levels of CXCL10 are expressed by intestinal ECs in AIDS patients with active infection and symptomatic disease [62]. *C. parvum* has also developed several strategies to evade the host immune response. For example, the chemokine CCL20 with antimicrobial properties is involved in the recruitment of DCs, but its expression is downregulated in neonatal mice after *C. parvum* infection, independent of the microbiota or IFN-γ response [63].

Toll-like receptors are an evolutionarily conserved family of cell receptors that play a vital role in immune responses by recognizing invading pathogens, including parasites. Most TLRs, upon the recognition of pathogen-associated molecular structures, use a common adaptor protein known as myeloid differentiation protein 88 (MyD88), leading to the activation of transcription factors NF-κB and the production of proinflammatory cytokines [64,65]. The production of T-helper type-1 (Th1) cytokines by *C. parvum*-treated mouse DCs is MyD88-dependent, and the bone-marrow-derived cells (BMDCs) derived from MyD88 KO mice fail to respond to *C. parvum* infection [66]. Consistent with this finding, MyD88 KO mice infected with *C. parvum* showed increased susceptibility to infection [67]. *C. parvum* infection in mice has been reported to increase TLR4 expression, suggesting it plays a role in activating DCs to detect *C. parvum* and initiate Th1 immune responses to restrict parasite growth [66]. The importance of TLR4 signaling in DCs was further established using BMDCs from mice lacking functional TLR4 signaling pathways, which failed to produce IL-12 upon *C. parvum* infection [66,68]. Consistent with this observation, TLR4 signaling was found to be crucial for the clearance of *C. parvum* infection in human biliary ECs known as cholangiocytes [31,69].

Despite the above evidence implicating TLR4 in *Cryptosporidium* infection, there are several alternative models that could explain these findings. TLR4 recognizes bacterial outer membrane component lipopolysaccharide (LPS) [70]. The enzymes responsible for endotoxin synthesis are absent in the *C. parvum* genome; hence, it does not express LPS or LPS-like molecules. However, parasite antigens such as *Toxoplasma gondii* (*T. gondii*) and Plasmodium glycosylphosphatidylinositol (GPI) have also been described as ligands of TLR4 [70,71,72]. Hence, a variety of glycoprotein antigens on the surface of *C. parvum* could also act as TLR4 ligands, although none have been definitively shown to act in this manner. The signal might also arise from contaminants in the oocyst preparation, as they come from fecal material that is rich in pathogen-associated molecular patterns (PAMPs). The inclusion of controls such as heat-inactivated oocysts could help resolve this issue in future studies. Infection with *C. parvum* also results in the increased intestinal permeability of neonatal mice, resulting in a leaky gut, thereby exposing the gut mucosa to commensal bacteria [73]. Hence, the microbial antigens derived from these bacteria could also act as pathogen recognition receptor ligands and may provide an additional signal to DCs. Further work is required to confirm the role of TLR4 and other TLRs in the recognition of *C. parvum* and the induction of downstream signaling pathways in DCs.

During *C. parvum* infection, DCs function as a first line of defense and play a critical role in activating immune responses. The depletion of DCs in adult mice is accompanied by a significant increase in oocyst shedding and more intestinal pathology [74]. The adoptive transfer of DCs stimulated with live parasites in vitro prior to transfer resulted in reduced parasite burden in mice with longer protection, highlighting the key role of DCs in controlling *C. parvum* infection [74]. Neonatal mice are more susceptible to *C. parvum* infection compared with adults due to the fact they have less well-developed populations of cDC1 cells. Intestinal DCs, especially CD103+ CD11b- DCs, are one of the key players in the innate immune control of *C. parvum* infection in neonatal mice [61]. One of the factors responsible for the higher susceptibility of neonates to *C. parvum* infection is the lesser recruitment of intestinal CD103+ DCs during the first few weeks of life due to weaker chemokine production by intestinal ECs [61]. The Flt3L ligand promotes the differentiation of DCs, and the administration of Flt3L to neonatal mice results in an increase in the DC population and decreased susceptibility to infection [61]. The development of a specific subset of DCs, called cDC1, revealed that the lineage of conventional dendritic cells is dependent on the transcription factor Batf3 [56,75]. Batf3-dependent cDC1 cells provide resistance to *Cryptosporidium* infection during the acute phase via cytokine production, and they are required for the clearance of the parasite in neonatal mice [76]. Upon infection, cDC1 cells are specialized to produce cytokines such as IL-12 and IFN-γ, which further induces IFN-γ production in immune cells such as ILC1, NK, T cells, DCs, and macrophages for the efficient control of *Cryptosporidium* infection [61,76]. 

After recruitment to the site of infection, DCs are known to capture antigens, process them, and then migrate to secondary lymphoid organs via the circulatory system to present the antigens to T cells and initiate an adaptive immune response [77,78]. One of the early studies in guinea pigs reported that macrophages phagocytose free *C. parvum* and transfer them further to DCs for migration [79]. DCs present in the lamina propria have dendrites that protrude into the gut lumen to acquire microbial antigens [80]. Further research in mice revealed that *C. parvum* could actively invade DCs that extend into the lumen of the intestine and be transported to mesenteric LNs [66]. A previous experiment revealed that both parasite antigens and live parasites are transported to lymph nodes [66]. It is not known how DCs engulf *C. parvum* antigens. This process could potentially occur from the direct parasite capture by transepithelial dendrites, the capture of apoptotic infected intestinal ECs containing parasites, or antigens secreted across the parasitophorous vacuolar membrane followed by DC uptake. Infection studies using transgenic *Cryptosporidium* parasites expressing fluorescent reporter proteins combined with live cell imaging could help explore these mechanisms.

## 4. Outcome of Dendritic Cell Interactions with *Cryptosporidium*

Dendritic cells secrete numerous cytokines, including IL-1β, IL-6, IL-12, IL-18, and TNF-α, upon *C. parvum* infection [81]. Both *C. parvum* sporozoite and recombinant antigens have been shown to not only activate murine DCs but also human DCs to produce cytokines [81]. Additionally, DCs derived from murine bone marrow cells express IFN-α and IFN-β in vitro after exposure to live parasites [35]. Gut flora also synergizes the responses of DCs through the activation of TLR5–MyD88 signaling for the optimal production of IFN-α and IFN-β. Consistent with this, treatment with poly(I:C) is necessary for controlling infection in neonatal mice [82]. During *C. parvum* infection, IL-12 is a key cytokine involved in promoting Th1 responses [83,84]. IL-12KO mice are more susceptible to infection, and the treatment of both immunocompetent and immunodeficient mice with IL-12 prior to *C. parvum* infection greatly limits infection severity [83]. IL-12 is an important regulator of IFN-γ and controls *C. parvum* infection in an IFN-γ-dependent manner [83].

*Cryptosporidium tyzzeri* naturally infects wild and laboratory mice, although different variants cause persistent [46] vs. short-term infection [85]. Using these models, the new role of DCs has been explored in the murine system. Intestinal ECs produce IL-18 upon NLRP6 inflammasome activation, and together with IL-12 produced by DCs, they induce early IFN-γ production by innate lymphoid cells in mice [37,85]. IFN-γ then activates the transcription factor STAT1, specifically in the intestinal ECs to restrict the growth of the parasite [37]. This work demonstrates that DCs provide an essential component of mucosal defense against intracellular parasites. We recently described a strain of *C. tyzzeri* named Ct-STL, which is a normal part of the endogenous flora of laboratory mice as a non-pathogenic commensal that establishes a long-term infection and only presents as a pathogen in cDC1-deficient mice [46]. The Ct-STL strain is capable of colonizing mice for many years, without having any observable clinical signs or symptoms in wild-type mice. Importantly, Ct-STL colonization induced cDC1 cells to produce IL-12 and expand IFN-γ-producing Th1 cells but not T-helper 17 cells (Th17) or regulatory T cells (Tregs). It is possible that, upon encountering the parasite, cDC1 cells provide an IL-12 signal that prevents the generation of Tregs by other cDC subsets. In contrast, mice deficient in cDC1 cells succumb to infection due to the reduced production of cytokines such as IL-12 and IFN-γ, compromised Th1 immune response, and skewed T-cell development toward Th17 and Tregs. However, it remains unknown whether these skewed CD4 T-cell responses and reduced IL-12 in cDC1-deficient mice cause fulminant cryptosporidiosis. Consistent with human studies, colonization by Ct-STL is mainly confined to the small intestine in mice. Interestingly, chronic infection with Ct-STL has an immunological impact in distal sites such as the colon, resulting in potent Th1 responses, elevated proinflammatory cytokines such as IL-12 and RANTES, and altered microbiome composition. These changes affect the function of the immune cells within the colon and lead to more resistance to *Citrobacter rodentium* infection and more susceptibility to DSS-mediated colitis in mice [46]. These findings imply that persistent infection with *Cryptosporidium* might alter responses to other intestinal infections or agents that trigger inflammation. The Ct-STL strain has uncovered the important role of dendritic cells during *Cryptosporidium* infection, and it should be a valuable tool for future studies. A similar phenomenon has been described following chronic *T. gondii* infection [86], although it is uncertain if it arises due to a common mechanism.

## 5. Effect of Microbial Dysbiosis on Dendritic Cells during Infection

The microbiota plays a critical role in the training and development of intestinal immunity [87]. Neonatal mice are much more susceptible to *Cryptosporidium* infection than adult mice [88]. The difference between adult and neonate microbiota or metabolites enriched in the neonatal gut beneficial to the growth of the parasite is suspected to be involved in this difference in susceptibility [82,89]. The lack of exposure to microbiota due to antibiotic use or exposure to dysbiotic microbiota during early life can lead to increased susceptibility to inflammatory diseases later in life [90,91]. Consistent with this model, mice exposed to *C. parvum* at an early age were found to be more susceptible to Salmonella infection than adults, thus raising the possibility that altered microbiota may play a role in the detrimental effect of cryptosporidium infection early in life [92]. Studies involving the examination of the microbial community in neonatal mice, infants, and young calves identified relationships between *Cryptosporidium* infection and microbial dysbiosis [93,94,95]. In young children, microbiome composition is predictive of *Cryptosporidium*-associated diarrheal symptoms at the time of infection [94]. The absence or low abundance of Megasphaera is more common in *Cryptosporidium*-associated diarrhea, while a higher abundance is associated with subclinical infection [94]. Megasphaera are gram negative bacteria, part of *Firmicutes* genus and classified within the class Clostridia. Megasphaera is known to synthesize short-chain fatty acids (SCFAs), which help maintain the homeostasis of gut health [96,97]. SCFAs such as acetate, propionate, and butyrate are known to have inhibitory effects in HCT-8s against *C. parvum* [98]. Interestingly, in HCT-8s, medium or long-chain saturated fatty acids also inhibit *C. parvum* growth, while long-chain unsaturated fatty acids enhance parasite invasion [89]. In the neonatal mouse gut, *Cryptosporidium* infection impacts the intestinal microbiota composition, resulting in an increased prevalence of bacterial communities belonging to the Phylum Bacteroidetes [93]. During the neonatal period, a special subset of Tregs is formed in response to the microbiota that maintains homeostasis and suppresses inflammatory responses; therefore, a healthy microbiota is required at the earliest steps in Treg priming [99]. A recent study reported that DCs from antibiotic-treated neonatal mice are incapable of generating a Treg population, which predisposes them to inappropriate immune responses [100,101]. Hence, microbial dysbiosis due to cryptosporidiosis could impact the DC-dependent Treg priming in the mesentery lymph nodes, resulting in excessive effector T-cell responses that lead to defects in long-term gut health and susceptibility to enteric infections. Future studies should be performed to investigate the association between microbial dysbiosis and DC-mediated immune responses following *Cryptosporidium* infection in early life.

## 6. Conclusions and Future Directions

The use of mouse-adapted strains *C. parvum* and mouse-specific *C. tyzzeri* have provided tractable systems to study the host immune response to this parasite. *C. parvum* does not infect laboratory mice well; in contrast, *C. tyzzeri* naturally infects wild and laboratory mice, although different variants cause persistent [46] vs. short-term infection [85]. Using these models, the role of DCs has been explored in the murine system. Further work is required in understanding the role of DCs in the activation of the T cells and exploring molecular mechanisms and identifying PAMPs involved in the activation of the TLR signaling in DCs during infection. Animal models have been important for investigating the immune response against this parasite; however, immune responses often differ between mice and humans; therefore, it is important to compare studies in both human and mouse models. The recent development of in vitro culture models including 2D and 3D organoid/enteroid culture models [102,103,104] and the ability to genetically manipulate *Cryptosporidium* will greatly help to further investigate host–pathogen interactions. DCs contribute to the host immune response against *Cryptosporidium* by stimulating innate and adaptive immune mechanisms. Studies have shown DCs as natural adjuvants in different infectious disease models to stimulate the immune response against various types of pathogens [105]. Clinical trials using antigen-pulsed DCs have confirmed the increased survival rate of vaccinated cancer patients [106,107]. Therefore, future studies should explore the role of DCs to enhance immune responses for immunotherapy or to develop a DC-based vaccination approach against *Cryptosporidium.*

## Figures and Tables

**Figure 1 microorganisms-11-01056-f001:**
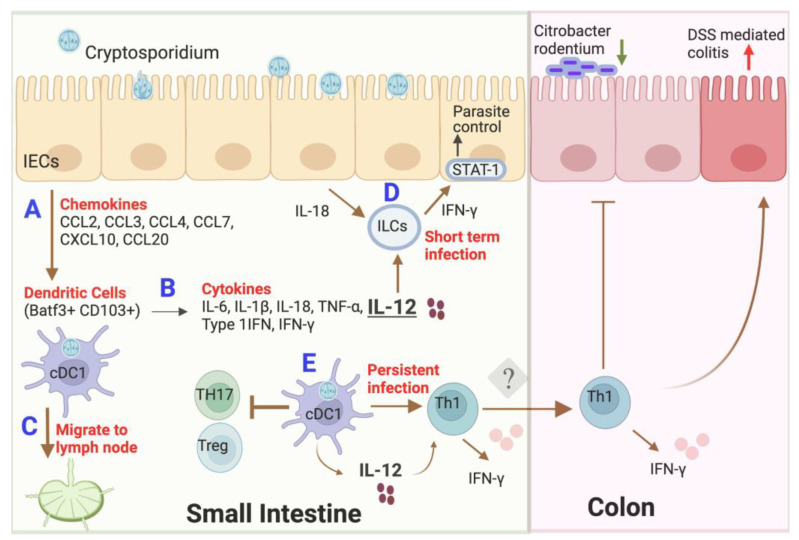
Interaction of dendritic cells with *Cryptosporidium*: (**A**) in response to *C. parvum,* IECs secrete a broad range of chemokines to recruit DCs at the site of infection; (**B**) Batf3+CD103+ DCs (cDC1) are recruited to the ileum and secrete numerous cytokines via TLR4 receptor activation; (**C**) DCs capture *C. parvum* antigens and live parasites in the gut mucosa and migrate to draining lymph nodes, where they present these antigens to naïve T cells and facilitate the adaptive immune response; (**D**) IL-18 from IECs works together with IL-12 produced by dendritic cells to induce type-I innate lymphoid cells to secrete IFN-γ. Then, IFN-γ activates the transcription factor STAT1 specifically in the intestinal epithelial cells to control the parasitic infection (short-term infection); (**E**) colonization by Ct-STL strain induces a cDC1-dependent antigen-specific Th1 response in mice that suppresses cryptosporidiosis. Although Ct-STL predominantly colonizes the small intestine, Ct-STL-specific Th1 cells and inflammatory cytokines are upregulated in the colon by a yet unknown mechanism. This persistent Th1 profile decreases susceptibility to *Citrobacter* infection but exacerbates DSS-induced colitis in the colon (persistent infection). Abbreviations: IECs, intestinal epithelial cells; IL, interleukin; TNF, tumor necrosis factor; IFN, interferon; ILCs, innate lymphoid cells; Th1, T helper type 1; Th17, T helper 17 cells; Tregs, regulatory T cells; DSS, dextran sulfate sodium. Created with BioRender.com.

**Table 1 microorganisms-11-01056-t001:** Dendritic cells’ function and subsets.

Dendritic Cell Subset	Conventional Type-I (cDC1).	Conventional Type-II (cDC2)	Plasmacytoid DCs (pDCs)	Monocyte Derived DC (Mo-DCs)
Transcription factor	BATF3, ID2, IRF8, ZFBTB46	IRF2, IRF4, RelB, ZFBTB46	E2-2, IRF4, IRF7	KLF4, IRF4,
Key markers for identification	CD11c^+^, MHC-II^+^, FLT3^+^, CD11b^–^, CD8a^+^, CD103^+^, XCR1^+^	CD11c^+,^ MHC-II^+^, CD24^+^ CD135^+^	CD11c^low^, MHC-II^int^ B220^+^ PDCA1^+^ Siglec H^+^, Lyc6C^+^	CD11b^+^ Lyc6C^hi/lo^, CD64^+^, CD14^+^
Functions	Act against intracellular pathogens IL-12 production; cross-presentation to CD8 T cells (MHC Class I)	Act against extracellular pathogens IL-12, IL-23, and TNF-α production Presentation to CD4 T cells (MHC Class II)	Antiviral response-type-I and type-III IFNs against viral infection. TLR7 and TLR9 induced CD4 and CD8 T-cell response	Differentiates from monocytes during inflammation with dendritic cell morphology. TNF-α, iNOS, and ROI production; Presentation to CD4 T cells (MHC Class II)

Abbreviations: BATF3, basic leucine zipper ATF-like transcription factor; IRF8, interferon regulatory factor; ZFBTB46, zinc finger and BTB domain containing; KLF4, Krüppel-like factor 4; iNOS, inducible nitric oxide synthase; ROI, reactive oxygen intermediates.

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
