# Peer review of "Dendritic Cells and Cryptosporidium: From Recognition to Restriction"

_microorganisms, 2023, doi:10.3390/microorganisms11041056_

Round 1

Reviewer 1 Report

This is an very interesting review article and I have only a few minor comments.

The genus name should be italicized even when written without the species. It is not consistently italicized throughout the text and is not capitalized on occasion.

Line 41. The beginning of this sentence “several species of Cryptosporidium exist…”  is unclear as the cited authors stated in that reference that there are over 38 species of Cryptosporidium. A more recent review from those authors states that there are 44 species and >120 genotypes.

Line 60: There is an error in the number of cases of cryptosporidiosis cases written. Gharpure et al reported that the 444 cryptosporidiosis outbreaks resulted in 7,465 cases.

Reference 17 is placed following a statement of cryptosporidiosis cases in the USA from 2009-2017; however, reference 17 was published in 2002.

There are many abbreviations throughout the text and it would help readers who are less familiar with immunology to provide more full definitions at first use, such as on Line 291 for T helper cells (T), and  Line 296  for T regulatory cells (Treg). 

The image in Figure 1 while interesting, is a little blurry, and I suggest you move it from page 6 to page 5 to be closer to the text description.

Lines 168 and 284 start the same way, “Cryptosporidium parvum does not infect laboratory mice well, …  I suggest rephrasing the start of the sentence on Line 284 before talking about the C. viatorum model.

Line 328. Please provide a very brief description of Megasphaera.

Author Response

Reviewer #1

This is an very interesting review article and I have only a few minor comments.

The genus name should be italicized even when written without the species. It is not consistently italicized throughout the text and is not capitalized on occasion.

Response- Thanks for pointing this out. We have made the correction throughout the text.

Line 41. The beginning of this sentence “several species of Cryptosporidium exist…”  is unclear as the cited authors stated in that reference that there are over 38 species of Cryptosporidium. A more recent review from those authors states that there are 44 species and >120 genotypes.

Response- We have added the recent review by Ryan et al, 2021 as suggested.

Line 60: There is an error in the number of cases of cryptosporidiosis cases written. Gharpure et al reported that the 444 cryptosporidiosis outbreaks resulted in 7,465 cases.

Response- This sentence is no longer contained in the revised manuscript. As suggested by reviewer #2, we have reduced the Cryptosporidium introduction section.

Reference 17 is placed following a statement of cryptosporidiosis cases in the USA from 2009-2017; however, reference 17 was published in 2002.

Response- This sentence is no longer contained in the revised manuscript. As suggested by reviewer #2, we have reduced the Cryptosporidium introduction section.

There are many abbreviations throughout the text and it would help readers who are less familiar with immunology to provide more full definitions at first use, such as on Line 291 for T helper cells (T), and  Line 296  for T regulatory cells (Treg). 

Response- We have included definitions for Th1, Th17 and Tregs.

The image in Figure 1 while interesting, is a little blurry, and I suggest you move it from page 6 to page 5 to be closer to the text description.

Response- We agree with this and have now added a higher resolution figure. We have also moved the Figure to page 5.

Lines 168 and 284 start the same way, “Cryptosporidium parvum does not infect laboratory mice well, …  I suggest rephrasing the start of the sentence on Line 284 before talking about the C. viatorum model.

Response- Thanks for pointing this out. We have removed it from Line 284.

Line 328. Please provide a very brief description of Megasphaera.

Response- We have added a statement to briefly describe Megasphaera.

Reviewer 2 Report

This is a review on the importance/role of DC's in the immune response to infection by Cryptosporidium. The review is well written and provides very useful information to the reader, however, the new advances or new knowledge regarding the role of DC's in the intestinal mucosa in cryptosporidium infections, which this review attempts to put into context with the already established knowledge, seem a bit lost in the manuscript. Extensive revision of the manuscript is not necessary, but the paragraphs need to be re-structured somewhat to put a stronger emphasis on new knowledge and on the role of the DC's.  Also, the introductory remarks on cryptosporidium itself could be shorter. 

Author Response

This is a review on the importance/role of DC's in the immune response to infection by Cryptosporidium. The review is well written and provides very useful information to the reader, however, the new advances or new knowledge regarding the role of DC's in the intestinal mucosa in cryptosporidium infections, which this review attempts to put into context with the already established knowledge, seem a bit lost in the manuscript. Extensive revision of the manuscript is not necessary, but the paragraphs need to be re-structured somewhat to put a stronger emphasis on new knowledge and on the role of the DC's.  Also, the introductory remarks on cryptosporidium itself could be shorter. 

Response- We agree with this and have now revised the text to highlight the newer studies. We have also reduced the Cryptosporidium introduction section.